# [Re] Temporal Spike Sequence Learning via Backpropagation for Deep Spiking Neural Networks

1 ## Reproducibility Summary

2 **Scope of Reproducibility**

3 In this report, we reproduce the results of a novel learning method for Spiking Neural Networks (SNN) proposed by
4 Zhang and Li (2020) [23]. The proposed learning method utilises biologically more plausible neuron interactions
5 than existing SNN algorithms. The original paper claims that the method can produce higher performance than other
6 state-of-the-art SNN learning algorithms on ML benchmarking datasets whilst utilising many fewer timesteps. In order
7 to test this claim, we reproduced the results of two datasets; MNIST and CIFAR-10.

8 **Methodology**

9 For the reproduction of the experiments in the paper [23], we used the author's original source code with minimal
10 additions; logging facilities and plotting functionality. We also performed an additional hyperparameter search
11 experiment for the MNIST dataset. The experiments were run on two different GPUs, an NVIDIA Tesla V100-PCIE-
12 32GB GPU and an NVIDIA Titan RTX. The total GPU runtimes were 150h 26m and 56h 4m, respectively. Additional
13 to the experiments performed, we inspected the theoretical equations in the original paper. We then scrutinised the
14 source code and the specific implementations of the mathematical equations.

15 **Results**

16 Due to high computational requirements, we reproduced two out of four experiments from the original paper. Overall,
17 the results match within a reasonable margin as reported in the paper. A Bayesian hyperparameter search, through
18 different combinations of parameters, revealed some insights about the stability and the speed of the training process.

19 **What Was Easy**

20 The original paper is well-written, with clear explanations of the models and the learning algorithm. We also thank the
21 authors for publishing their source code online. This made the reproduction study easier and more fruitful. The source
22 code was written in an understandable way and the authors provided clear general instructions to rerun the networks.

23 **What Was Difficult**

24 The computationally demanding nature of the networks yields it challenging for us to reproduce all the experiments in
25 the paper. Particularly, SNNs require multiple timesteps that leads proportionally longer runtime. Additionally, we
26 discovered some parameters that were hardcoded and undocumented in the source code without explanations. We could
27 not find their particular contexts in the original paper.

28 **Communication with Original Authors**

29 We contacted the authors on several occasions to ask questions about the paper and the undocumented parameters in the
30 source code. They kindly clarified all our questions. We also provided some feedback regarding theoretical equations
31 and code implementation.

# 1 Introduction

Spiking Neural Network (SNN) models are based on biological networks which utilise spikes as a method of information transmission. Spiking is highly energy efficient, meaning these networks provide attractive computational solutions [5]. Several neuromorphic chip hardware have been developed to run spiking networks [1, 3], however, neural network algorithms that fully utilise their capabilities have yet to be realised. Deep neural networks have received increasing interest over recent years, inspiring the development of efficient deep spiking algorithms (Deep-SNNs) that can be run over multiple layers [17]. Deep-SNNs vary in both spatial and temporal aspects, leading to complicated network dynamics. The utilisation of spiking codes for machine learning tasks is therefore nontrivial. This is specifically due to the challenges of employing backpropagation—the typical basis for calculating weight updates. Previous spiking network algorithms have solved the problem of non-differential discrete spike events via surrogate gradients or approximation using continuous activation functions. However, these techniques destroy crucial temporal aspects of SNNs—previous spikes of a neuron affect future spikes. In addition, the use of spikes in previous SNN models has required large temporal latency—a greater number of timesteps, in order to achieve more accurate performance on tasks. This makes scaling to deep network architectures with many layers computationally expensive. In this report, we review the proposed method of "temporal spike sequence learning via backpropagation" (known as TSSL-BP) which claims to deal with both of these problems by considering inter and intra-neuron spiking dependencies, and reducing the required number of timesteps.

Previous studies [19, 6, 15, 10, 7, 18, 16, 22] have demonstrated increasing accuracy of spike-based network algorithms on image classification datasets such as MNIST [9], Neuromorphic MNIST (N-MNIST) [12], Fashion-MNIST [20], and CIFAR-10 [8]. TSSL-BP demonstrates increased accuracy on all these datasets, including a 3.98% increase for the more challenging CIFAR-10. Critically, the network can not only perform at higher accuracy, but requires much shorter time-window.

## 1.1 TSSL-BP Overview

The aim of the TSSL-BP algorithm is to learn a desired firing sequence, that can be set arbitrarily. The error function for the network to be minimised is the distance between the produced spiking pattern and the desired sequence (target). In TSSL-BP, the loss is defined as the sum of the squared error over all neurons for each timestep. The distance between the actual and desired spiking times is

$$L = \sum_{k=0}^{N_t} E[t_k] = \frac{1}{2} \sum_{k=0}^{N_t} \left( (\epsilon * \mathbf{d})[t_k] - (\epsilon * \mathbf{s})[t_k] \right)^2, \tag{1}$$

where $E[t_k]$ is error at discrete timestep $t_k$, $\mathbf{d}$ and $\mathbf{s}$ are the desired and actual spike trains, and $\epsilon$ is a kernel function measuring the Van Rossum distance between them [13]. The spike trains are binary sequences within a certain time-window.

### 1.1.1 Leaky Integrate-and-Fire Model

Spikes in the neural network model are generated using the standard Leaky Integrate-and-Fire (LIF) model [4]. This model describes a neuron's membrane potential over time and generates a spike if the potential value is a higher than a specified threshold. Incoming spikes are converted into a postsynaptic current (PSC) $a_j(t)$. The neuronal membrane potential $u_i(t)$ for neuron $i$ is

$$\tau_m \frac{du_i}{dt} = -u_i(t) + R \sum_j w_{ij} a_j(t) + \eta_i(t), \tag{2}$$

where $R$ and $\tau_m$ are leaky resistance and time constant of the membrane, $w_{ij}$ is the synaptic weight between neurons $i$ and $j$, and $\eta_i$ is the reset function. The PSC and the reset function are defined as,

$$a_j(t) = (\epsilon * s_j)(t), \quad \eta_i(t) = (\nu * s_i)(t), \tag{3}$$

where $s_j$ is the spike times of neuron $j$, $\nu$ is reset kernel and $\epsilon$ is spike response. The spike response kernel is

$$\tau_s \frac{a_j}{dt} = -a_j(t) + s_j(t), \tag{4}$$

where $\tau_s$ is synaptic time constant.

Finally, the firing output is determined by the Heaviside step function, $H(.)$, producing all-or-none spiking depending on whether the membrane potential is over a specified threshold $V_{th}$:

$$s_i[t] = H(u_i[t] - V_{th}). \tag{5}$$

### 1.1.2 Inter and Intra-neuron Dependencies

Spiking outputs are binary; all-or-none discrete events, meaning the activation function is not differentiable. Some SNN implementations [18, 16, 21] have circumvented this issue via use of surrogate gradient methods [11]. However, these methods degrade training performance due to discrepancies between the target loss and gradient. More recent SNN methods have reduced the number of timesteps required, however, still utilise continuous activation functions as approximations of the spiking neurons [19]. The approach of TSSL-BP considers the precise temporal dependencies of both spiking between neurons (inter) and within neurons (intra). Inter-neuron dependencies arise as the spikes from the presynaptic neurons cause changes in the postsynaptic neurons' current. Furthermore, due to the membrane potential reset kernel in the LIF model, the timing of spikes within a neuron can be dependent on previous spikes within a certain time-window. This affects the precise timing of spikes and the subsequent postsynaptic current. We refer the reader to see the original paper for the derivation.

## 2   Scope of reproducibility

In this reproduction, we test the following claims of the paper:

1. TSSL-BP performs at a higher accuracy than previous spiking neural nets on supervised learning problems using the MNIST and CIFAR-10 datasets. A large increase of 3.98% on CIFAR-10 can be achieved, which is a challenging dataset for SNNs.

2. Compared to previous SNNs, TSSL-BP utilizes a lower latency of spikes, whilst still maintaining high accuracy on the datasets.

To test these claims, we first checked the mathematical derivations provided in the paper, supplementary material and the source code implementation. Second, we tested the claims about performance accuracy by reproducing the results for the TSSL-BP method on the MNIST and CIFAR-10 datasets. An additional hyperparameter search was conducted for the MNIST dataset to assess whether the parameters selected in the original paper were justifiable.

## 3   Methodology

### 3.1   Model Descriptions

The neural network architecture and the learning algorithm were implemented in PyTorch ML framework.[1] The network architecture is an adapted ConvNet, modified to allow spike generations. The network architecture used for MNIST dataset was 15C5-P2-40C5-P2-300, it has 210,375 trainable parameters in total. The CIFAR-10 dataset was tested on two different network architectures. The first one (CNN1) was 96C3-256C3-P2-384C3-P2-384C3-256C3-1024-1024 and the second (CNN2), larger network 128C3-256C3-P2-512C3-P2-1024C3-512C3-1024-512. For both, a fixed probability of 0.2 dropout was applied after each layer. The networks have 21,156,384 and 44,999,040 trainable parameters, respectively. Outputs of the model are converted to spike rates and tested against desired spike rates for each class. The target class is given a desired spike count, and other classes are set to target zero spikes.

At the beginning of network training, low initial weight values can lead to an absence of firing activity, meaning backpropagtion with TSSL-BP is not possible. To solve this, the authors suggests to apply a warm-up mechanism, where an average firing rate threshold is set for each layer. When an activity is above this threshold, TSSL-BP is applied. If the firing activity is very low the warm-up is applied. In this case, the activation function is approximated using a continuous sigmoid function of membrane potential, allowing backpropagation without spiking.

### 3.2   Datasets and Hyperparameters

MNIST is a prominent benchmarking dataset of handwritten digits for image recognition tasks. It includes 70,000 gray-scale input images of size of 28×28 for 10 classes. Amongst these, 60,000 images are used for training and 10,000 for testing. CIFAR-10 is another well-known, image recognition benchmarking dataset. It includes 60,000 colour input images of size 32×32 for 10 classes. Amongst these, 50,000 images are used for training and 10,000 for testing. For both the MNIST and CIFAR-10 datasets, preprocessing was performed as in the original paper.[2] For each image, short time-windows of real-valued spike currents are generated from pixel intensities.

---

[1]TSSL-BP implementation provided by the authors is available on Github: `https://github.com/stonezwr/TSSL-BP`

[2]These datasets are downloadable as part of PyTorch package.

For the reproduction of the experiments, we used the same parameters as in the original paper and source code provided by authors. For both MNIST and CIFAR-10 datasets, batch size was 50, $\tau_s$ (synaptic time constant) was 3, $\tau_m$ (membrane time constant) was 5, time-window was 5 (with desired count of 4 and undesired 1). For MNIST, the networks were trained for 100 epochs at a learning-rate of 0.0005, and for CIFAR-10, 150 epochs at a learning-rate of 0.0002. An additional Bayesian hyperparameter search (12 runs) was performed for the MNIST dataset. The details of this experiment can be found in Section 4.3.

### 3.3 Experimental Setup and Computational Requirements

As in the original paper, we ran five trials each of the MNIST network and `CNN1` network (CIFAR-10 dataset). Due to runtime limitations, `CNN2` network was trained twice and the best performing is reported here. All performance data were measured by accuracy.

Our additional code is available online.[3] Weights and Biases API was implemented on the model code to track model learning and assist with analysis [2]. Reproduction of the results for MNIST and CIFAR-10 datasets were run on a NVIDIA Tesla V100-PCIE-32GB GPU, and the hyperparameter search for MNIST dataset was run on a NVIDIA Titan RTX. The detailed GPU runtimes are:

- For the MNIST dataset, the mean run time was 1h 21m and the total GPU time for 5 runs was 6h 47m.
- For the CIFAR-10 dataset, the mean run time for `CNN1` network was 18h 33m hours and the total GPU time for 5 runs was 92h 43m.
- For the CIFAR-10 dataset, the mean run time for `CNN2` network was 25h 28m and the total GPU time for 2 runs was 50h 55m.
- For the MNIST dataset hyperparameter search, the total runtime of 12 runs was 56h 4m.

## 4  Results

This section reports reproduction of two experiments from the paper; MNIST and CIFAR-10, and an additional hyperparameter search for MNIST. Overall, our results supports the claims in the original paper. Accuracy scores for each network reproduction were within reasonable margin of the original paper.

### 4.1  Result 1: MNIST

We conducted five runs for MNIST dataset, with the same hyperparameters as the original paper. Our reproduction produced a mean accuracy of 99.40% (see Table 1). Compared to the paper, this was within reasonable margin, original 99.50% with ours 0.1% lower. Between the best performance of each, the difference was only 0.06%.

Table 1: Performances comparison of the original paper and our reproduction for MNIST dataset.

| Method | Network | Mean Accuracy | Std. Deviation | Best Performance |
|---|---|---|---|---|
| Original paper | 15C5-P2-40C5-P2-300 | 99.50% | 0.02% | 99.53% |
| Reproduction | 15C5-P2-40C5-P2-300 | 99.40% | 0.04% | 99.47% |

For the MNIST dataset, we confirm that TSSL-BP outperforms most other SNNs [7, 10, 16, 18]. As in the original paper, it performs marginally below ST-RSBP network [22]. ST-RSBP achieves 99.57%, versus 99.40% for our replication and 99.50% reported in the original paper. However, ST-RSBP with the same network architecture requires 400 timesteps, versus only 5 for TSSL-BP. Given this, the performance for TSSL-BP is highly comparable to other SNNs. This result also provides support for the claim that TSSL-BP can perform well even with few time steps.

### 4.2  Result 2: CIFAR-10

For CIFAR-10, the results of our replication can be found in Table 2. For the smaller CNN1 network, the original paper demonstrates best accuracy increases of 3.98% over STBP algorithm [19]. For our reproduction, we achieved mean accuracy of 88.96% versus 88.98% in the original paper. Best performance for the reproduction was slightly lower, 89.07% versus 89.22%. Compared to STBP, the best performance increase is 3.98% original and 3.83% reproduction.

---

[3]Github repository with our additional code: `https://github.com/anilozdemir/TSSL-BP`

Table 2: Performance comparison of the original paper and our reproduction for CIFAR-10.

| Method | Network | Mean Accuracy | Std. Deviation | Best Performance |
|--------|---------|---------------|----------------|------------------|
| Original paper | CNN1 | 88.98% | 0.27% | 89.22% |
| Reproduction | CNN1 | 88.96% | 0.10% | 89.07% |
| Original paper | CNN2 | - | - | 91.41% |
| Reproduction | CNN2 | - | - | 89.61 % |

CNN1: 96C3-256C3-P2-384C3-P2-384C3-256C3-1024-1024
CNN2: 128C3-256C3-P2-512C3-P2-1024C3-512C3-1024-512

In the original paper, it is unclear how many times the CNN2 architecture was run. We contacted the authors in that regard and they clarified that the network was run only once. The accuracy reported in the original work is 91.41%, a 0.88% increase on STBP (with NeuNorm) [19]. In this reproduction, we ran the network twice and selected the highest accuracy of the two, 89.61% versus 89.53%. This is a lower performance than STBP, with a reduction of 0.92%. It is unclear whether TSSL-BP would consistently score lower over more trials, or whether the lower score obtained here was due to network stochasticity. Nevertheless, TSSL-BP utilises marginally fewer timesteps, with a reduction from eight for STBP (with NeuNorm) to five. The authors also report that there are no additional optimisations on TSSL-BP that are used in the comparable SNNs [19], such as neuron normalisation and population decoding.

## 4.3 Results Beyond Original Paper

We investigated the hyperparameters used in the MNIST network, the original parameters can be found in Table 3. As it is costly to run the network and impractical to search for large numbers of parameters, we performed a brief Bayesian hyperparameter search for learning-rate, number of epochs and time-window. We utilised the sweep functionality from Weight & Biases ML developer tools.[4] For this search, we used Bayesian optimisation with the objective function improving the test-accuracy. We ran this optimisation 12 times, with parameters to be selected from:

- epochs $\in \{50, 100, 150, 200\}$
- learning-rate $\in \{0.0001, 0.0005, 0.001, 0.005, 0.01\}$
- time-window $\in \{5, 10, 20\}$

Table 3: Original hyperparameters used for the MNIST dataset.

| Parameter | Value | Parameter | Value |
|-----------|-------|-----------|-------|
| epochs | 100 | learning-rate | 0.0005 |
| batch size | 50 | time-window | 5 |
| desired count | 4 | undesired count | 1 |
| $\tau_m$ | 5 | $\tau_s$ | 3 |

Figure 1 shows an overview of the different parameters used and the corresponding test-accuracy after completing training. Within the limited number of runs, one can see that a longer time-window leads to higher accuracy. Although, there is a trade-off—the longer time-window increases runtime, due to longer input processing. On the other hand, the Bayesian optimisation method was favouring longer time-windows.

Figure 2 shows training accuracy obtained at each epoch for 12 runs. We observed sudden changes in three of the runs during training. This may be due to an instability of the learning algorithm—this could be explored in future work. Surprisingly, on the other hand, the test performance did not change. Amongst the unexpected runs, one had a peculiar learning process (see purple curve). The training performance accuracy periodically changed. Another observation is that some of the runs (e.g. top orange curve) increased to their maximum performance quite early on (less than 25 epochs) and the performance did not improve thereafter, suggesting that the number of epochs may have been set too high.

Figure 3 demonstrates the distribution of learning-rates against number of epochs, and the colours represent test-accuracy. It is clear that the effect of epoch is not substantial, though, learning-rate impacts greatly; lower learning-rate leads to better performance. Results suggest that approximately the same performance could be achieved with half

---

[4]Documentation for sweep function: https://wandb.ai/site/sweeps.

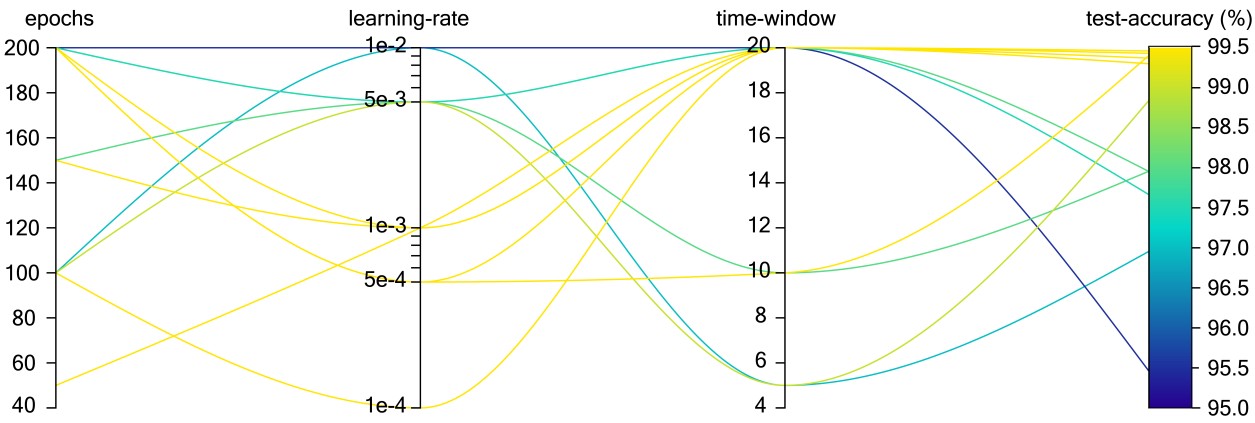

Figure 1: Parallel coordinates plot showing different combinations of hyperparameters and the resulting test-accuracy. The colours indicate the performance accuracy—the lighter the colour the higher the test-accuracy. Note that learning-rate is in log-scale.

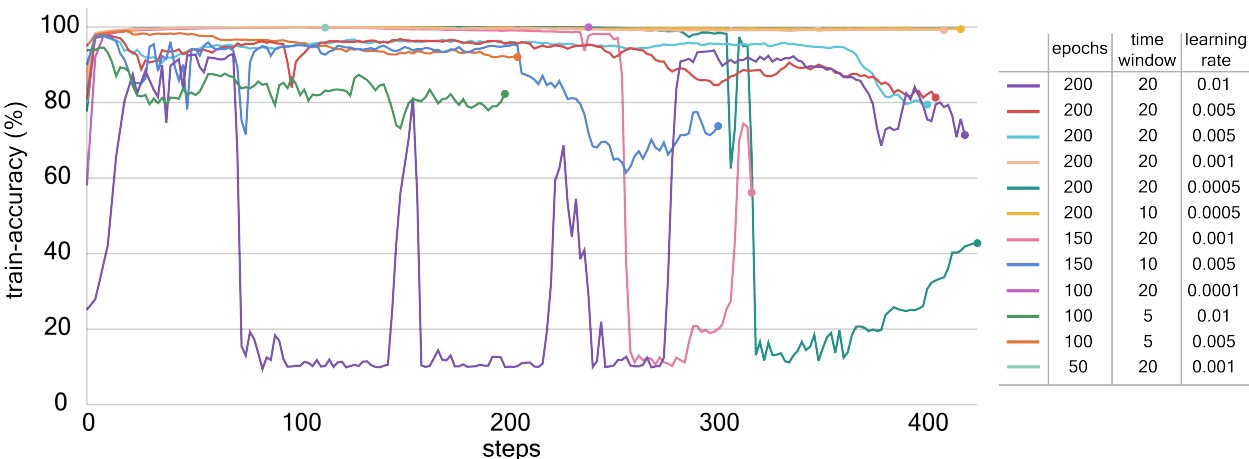

Figure 2: Training accuracy at each step for 12 runs. The hyper-parameter settings for each run are given in the legend. Note that horizontal axis shows the training iteration steps rather than the number of epochs. This is due to using W&B API and saving the results asynchronously. Overall trends in the plot, however, remain the same.

the amount of epochs, if the learning-rate is chosen appropriately, e.g. 0.001. For this learning-rate setting, three runs were performed. The test-accuracy results were 99.3%, 99.43% and 99.37% for number of epochs 50, 150 and 200, respectively. On the other hand, the time it takes to run each of them was 1h 50m, 5h 32m and 7h 23m. From this, there is no significant benefit of running the MNIST experiment for a large number of epochs.

Overall, the highest performance was 99.46%, using time-window of 20 for 200 epochs at learning-rate of 0.0005, however, this took 7h 22m. The lowest computational cost was 1h when using time-window of 5 for 100 epochs at learning-rate of 0.005 with a performance of 98.85%. Comparing this experiment with the paper, the original hyperparameters selected in the paper were well-optimised.

## 5 Discussion

Overall, the reproduction study was fairly straightforward. The authors were helpful and provided clear explanations. Due to time constraints and limited resources available, we could only reproduce two out of the four experiments from the paper. However, our choice of the reproduced experiments was deliberate; we chose a well-known and relatively simpler dataset (MNIST) and another well-known but more complicated dataset (CIFAR-10). The reproduced accuracy results were within reasonable ranges to the authors' original paper results. For both MNIST and CIFAR-10 (using a

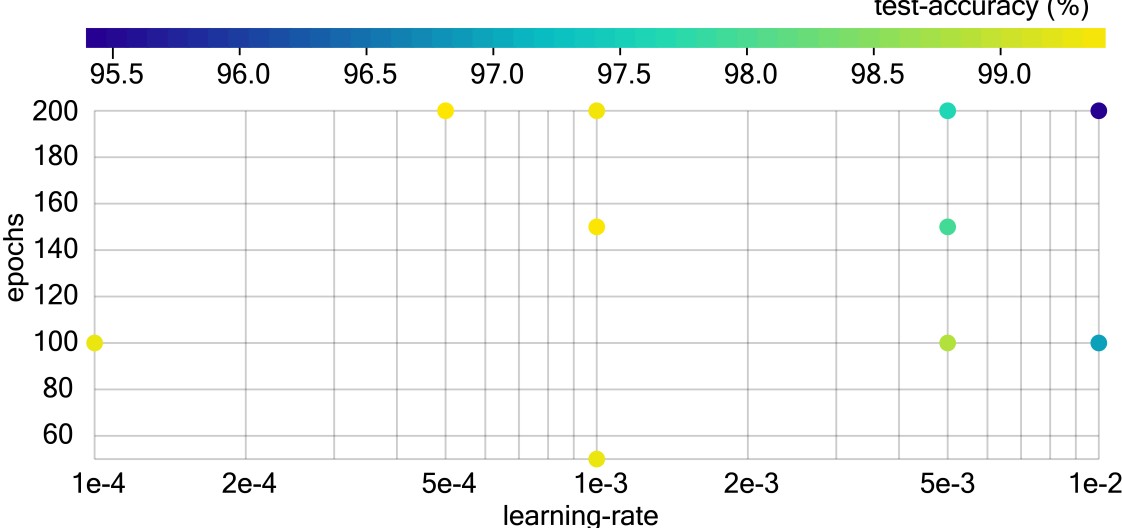

Figure 3: Distribution of 12 runs with respect to number of epochs and learning-rate, with colours indicating the test accuracy.

smaller network structure) TSSL-BP outperforms most other SNN algorithms. Furthermore, we found that the reduced number of timesteps is sufficient to reproduce these accuracies, supporting claim two.

We performed an additional hyperparameter search to investigate the proposed learning algorithm's abilities outside of the selected parameter domain. This investigation revealed some interesting properties; such as that comparable performance can be achieved in a shorter runtime, and that the learning algorithm may have some instabilities. These insights can lead to further experimentation and perhaps further novel contributions in the future.

**What was easy** Authors provided the necessary source code for the learning algorithm and most network setups. This made the reproducibility study more fruitful. The only missing network setup was for `CNN2`, however, this can easily be reproduced by amending the layer sizes provided in the `CNN1` file. We used W&B API for logging the results and plotting facilities, this made the collaboration experience easier and allowed us to monitor the network runs asynchronously.

**What was difficult** Some parameters included in the original source code were undocumented. These are: `a = 0.2` (line 89 in `functions/tsslbp.py`), `th = 1/(4 * tau_s)` (line 56 in `functions/tsslbp.py`), and `theta = 1.1` (line 31-32 in `layers/pooling.py`). It was also unclear how the network weight clipping was determined (-8 and 8 for `line 90` at `tsslbp.py` and -4 and 4 for `weight_clipper` function in `layers/linear.py` and `layers/conv.py`).[5] We contacted the authors to clarify these; `a` and `th` are used as rescaling factors and `theta` was not used in the code (i.e. redundant). Moreover, the authors confirmed that the particular values are empirically found and manually tuned. Finally, the network warm-up mechanism could be cumbersome for more complicated datasets.

**Communication with original authors** We communicated with the author at the NeurIPS poster session, where our initial questions were answered surrounding the method, implementation and goals of the network. Following this we provided the authors with feedback on the details of some of the equations via email. We also enquired about the undocumented parameters given in the source code and the number of runs performed for the CIFAR10 dataset using `CNN-2` network. We thank the authors for their engagement with this process.

**Future Work** We attempted to test the algorithm on another neuromorphic dataset used to benchmark SNN—a dynamic vision sensor (DVS) version of the CIFAR-10. During initial testing we found that large computational resources are required, and therefore did not proceed. We have provided the code for preprocessing of the dataset (based on [14]) in our code repository for future works to utilise.

The `CNN2` network reproduction demonstrated lower accuracy than the STBP algorithm [18]. It is unclear whether this is due to variability of running a low number of trials or a more general trend. As fewer timesteps were used for

---

[5]These lines references are for the current state of the GitHub repository. The authors highlighted that the code is still under development for further optimisation, so these line references may change.

the TSSL-BP implementation when comparing, it is possible TSSL-BP would achieve higher accuracy with the same number of timesteps. Future work could investigate whether comparing in this case yields higher accuracy as claimed by the authors.

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
