# OpenReview forum: "[Re] Temporal Spike Sequence Learning via Backpropagationfor Deep Spiking Neural Networks"
_ML_Reproducibility_Challenge/2020 — Reject_

### Official Review · AnonReviewer2 · 2021-02-20
**Not all experiments were confirmed and there is no proper hyperparameter search performed**

**Rating:** 4
**Confidence:** 4

**Review:**

The authors of the reproducibility paper provide a nice summary of the original manuscript. However, there are some major issues and shortcomings that prevent this reproducibility study to be accepted for publication. I list them here:

- As authors of the reproducibility paper stated, only two out of four experiments were performed. Considering that they did not need to code anything (the code was provided by the authors of the original paper), reproducing as many experiments as possible should have been a priority.
- Even with the lack of time and computational requirements, instead of performing a full reproducibility of MNIST experiment (the simplest and the first experiment of the original paper), a full reproducibility of neuromorphic version of MNIST: N-MNIST woudl have been more appropriate. This was the second experiment in the original paper and it would have been considerably more interesting to see reproducibility results for it.
- The hyperparameter search was only performed for one of the four experiments. In addition, it was the simplest experiment and the hyperparameter search was rather brief and not exhaustive. There is no mention of feedback from the authors of the original manuscript on the hyperparameter range they experimented with. Since the code was already available, I believe this section should have been investigated more thoroughly.
- It seems that the hyperparameters used in the reproducibility study (as listed in lines 115 to 118) do not correspond to the hyperparameters used in the original paper (as listed in Section 2.1 of Supplementary Materials).
- The authors of the reproducibility study mention discovering hardcoded parameters in the source code that can not be explained. I think a list of such parameters should have been made available somewhere (if not in the paper itself, then at least on the github webpage they provided). Furthermore, there is no attempt to explain or even guess what these parameters do. I understand that just plain changing of these parameters and re-running the experiments would require additional time and resources, but that would have been in the best interest of this reproducibility study. In addition, the authors of this study do not mention attempting to clarify these with the authors of the original manuscript.

In the end I do believe that if authors of the reproducibility study were given more time and resources, they would have made a considerably better study. Unfortunately, this is not the case.

**Familiar With The Original Paper:**

I have read the original paper

**Reproducibility Summary:**

Report has summary

---

### Official Review · AnonReviewer1 · 2021-03-02

**Rating:** 7
**Confidence:** 3

**Review:**

Overall, this was really nicely put together.

Reproducibility Summary: Provided, and contains a nice summary of the presented reproducibility results.
Scope of reproducibility:
- Narrowed down to two datasets (MNIST and CIFAR-10), to deal with computational constraints. I think this is well justified given the computational complexity of the reproduced algorithms.

Code:
- Original code used, with minor additions.
- Mentions checking major mathematical sections for mistakes/bugs.

Communication with original authors:
- Clarification and recommendations for theory communicated to authors (who reciprocated).

Hyperparameter Search:
- Conducted a separate hyperparameter search on MNIST using bayesian optimization provided by WB ML dev tools. They did the search over 3 parameters (epochs, learning rate, time window), see results beyond the paper for more details.

Ablation Study:
- Not applicable as far as I can tell.

Discussion on results:
- Did a good job discussing the results.
- Provided average run-times and total compute used
- Justified the experiments reproduced well.
- Discussion is well written and clear.

I think you should report which parameters are hardcoded in the original code, and what values they are set to. Possibly linking them to the actual notation of the algorithm rather than what is in the code.

It might be nice to include a concrete set of recommendations, rather than listing what is wrong. For example, "Document all hard-coded parameters and give justification for their choice."

Results beyond the paper:
- A new hyperparameter search was conducted over the MNIST dataset
- Did a nice job uncovering some potential stability concerns for future work.

Overall organization and clarity:
Well written and clear.

Questions/Concerns:
Did you only do a single run for the hyperparameter search? Confidence intervals for these novel results (Figure 1 and Figure 3) would be a nice addition. Maybe you did 12 runs? But it seems you did 12 different configurations. There might be high variability in the settings.

In figure 2, which line corresponds to which settings? It might be nice to try and figure out how to visualize this information, as this would be useful for making inferences about hyperparameter selection.


**Familiar With The Original Paper:**

I have read the original paper

**Reproducibility Summary:**

Report has summary

---

### Official Review · AnonReviewer3 · 2021-03-02
**Nicely-conducted replication of the original result**

**Rating:** 9
**Confidence:** 3

**Review:**

In this submission, the authors reproduce two of the four experiments originally performed by Zhang and Li (2020).  Confirming the original result, the authors find that Temporal Spike Sequence Learning Backpropagation improves spiking neural networks' performance to near SOA levels with reduced training time. Additional experiments were conducted to examine the influence of hyperparameters were also performed; the authors report that training times reported in the original paper may be pessimistic.

I have no major concerns. There does appear to be a discrepancy between the original paper and replication in the CNN2/CFAIR-10 experiment. However, the authors note that they were only able to run the experiment twice, and this may reflect network stochasticity (though the delta seems rather large). The other experiments replicate nicely, and the rest of the manuscript also contains a brief explanation of spiking neural networks and the authors' rationale for choosing to replicate these two experiments, which is nice.

The authors apparently reviewed the source code (Line 208ff) and found some undocumented parameters. A slightly longer discussion of these (e.g., where they are, what they seem to do) might be helpful for future readers of both papers. However, I leave this entirely to the authors' discretion.

**Familiar With The Original Paper:**

I have read the original paper

**Reproducibility Summary:**

Report has summary

---

### Decision · Program_Chairs · 2021-03-31

**Decision:**

Reject

**Comment:**

Well-written report, however not good enough for recommendation to ReScience journal due to limited ablation study/discussions, given that the paper already re-uses the original code. Thus, the ACs are unable to recommend this paper.